# "My mother in-law forced my husband to divorce me": Experiences of women with infertility in Zamfara State of Nigeria

**Florence Naab**[1]*, **Yakubu Lawali**[2], **Ernestina S. Donkor**[3]

**1** Department of Maternal and Child Health, School of Nursing and Midwifery, College of Health Sciences, University of Ghana, Accra, Ghana, **2** Department of Nursing Science, College of Health Sciences, Usmanu DanFodiyo University Sokoto, Sokoto, Nigeria, **3** Department of Maternal and Child Health, School of Nursing and Midwifery, College of Health Sciences, University of Ghana, Accra, Ghana

* florencenaab@yahoo.com, fnaab@ug.edu.gh

**Data Availability Statement:** All data is in the manuscript and supporting information files. The 12 anonymized transcripts are available as supporting information files.

## Abstract

### Introduction

Women with infertility have different experiences that determine the quality of their psychosocial health. Cultural constructions of infertility in Africa have made the experience of infertility among African women more burdensome. Yet, little is known about the psychosocial experiences of women with infertility in Zamfara State of Nigeria. The purpose of this study was to explore the psychosocial experiences of women with infertility in Zamfara.

### Materials and methods

An exploratory qualitative design was used for this exploration. Individual in-depth interviews were conducted using a semi-structured interview guide. Ethical approval was received from the Institutional Review Board of the Nogouchi Memorial Institute for Medical Research in the University of Ghana. Women who were married and seeking treatment for infertility from a public hospital were recruited and interviewed. A total of 12 women were interviewed with each interview lasting 45 minutes. The interviews were audiotaped with permission from the participants, transcribed verbatim, and content analysed.

### Results

The findings revealed that psychologically, majority of the women had experienced anxiety, stress and depression as a result of their inability to get pregnant. Socially, the women suffered self and social isolation, social stigma, social pressure and marital problems.

### Conclusion

These women have psychosocial health problems that need the attention of health professionals to enhance their wellbeing.

**Funding:** The authors received no specific funding for this work.

**Competing interests:** The authors have declared that no competing interests exist.

## Introduction

Infertility is considered as one of the six maternal morbidities neglected within developing countries [1] and it is reported as being 16 times more frequent than maternal mortality [1]. It has also been established that African women with infertility have been subjected to domestic violence due to their inability to conceive [2]. Consequently, couples with infertility live in fear and anxiety, especially if they have been diagnosed medically and are undergoing treatment [3]. This feeling of fear and anxiety may cause conflict between the spouses, a decrease in self-esteem, a decrease in frequency of sexual intercourse, and the development of feelings of inadequacy. As a result, marriages are put under intense psychological pressure because of infertility [4,5]. Therefore, infertility can be a reason for marital instability and divorce [4,5]. Furthermore, domestic violence has been reported to be associated with the experience of infertility among women and the violence is exhibited in the form of psychological torture, verbal abuse, ridicule, physical abuse and deprivation [6]. Socially, infertility is known to be associated with frustration, pain, social ostracism, stigma, marital instability, and suicide [7].

In Africa, women with infertility are more culturally blamed than sterile or impotent men because it is perceived that only women experience infertility [8]. As compared to couples with infertility in western societies, couples with infertility in low income countries feel a deeper sense of guilt, shame, worthlessness and depression [9]. Research has shown that in these parts of the world, women suffer more of the consequences than their male counterparts [9,10]. For instance, the psycho-social experiences suffered by infertile couples in Rwanda are severe and similar to those reported in other resource-poor countries [11]. Many women feel that they shoulder a disproportionate share of the blame for infertility and by extension face greater social consequences for difficulties in conceiving than their male partners [12,13]. South African women admitted to intense emotions such as anger, profound sadness, bitterness, loneliness, depression and some of these women have confessed to suicidal thoughts [14]. Similarly, depression and stress were found to be more prevalent symptoms and high among women with infertility in Ghana [12,13,15]. Socially, African women are reported to suffer social stigma as a result of infertility [16,17,18]. In Ghana for instance, women are reported to have social stigma, social isolation, and are faced with marital strain and instability [12,17]. Consequently, there is a reported increased risk of insecure sexual behaviour of both men and women with infertility in Ghana, increasing their risk for contracting HIV, and other sexually transmitted infections due to sexual promiscuity [12].

Another problem among African couples with infertility is the pressure from family and friends. Significant others such as mother in-laws and other relatives of the male spouse are reported to be the major source of pressure on African women with infertility [8]. However, it is not clear if the lineage system practiced in many African societies is to blame for this kind of pressure. On the other hand, because children are perceived as a source of social security in old age [8], many couples especially women, struggle with the pressure to secure their status in the family and in their old age. Friends of the couple may pass oblivious comments that are reported to be a source of pressure to conceive [8]. Unfortunately, such comments are usually more directed to the woman than the man, increasing the possibilities of severer mental health problems among women.

In Nigeria, the prevalence of psychiatric morbidities in general is reported to be high among women with infertility [18]. Among these psychiatry morbidities, depression and anxiety are significantly higher among women who are divorced as a result of primary infertility [18]. Furthermore, married women in Nigeria are only recognized when they have children, especially male children [19] for the continuity of the family lineage. As reported by these authors [19], the experience of infertility in Nigeria is culturally constructed to be a woman's

problem instead of a problem for the couple. These cultural constructions form various sources of infertility related stigma, often experienced by the woman. Consequently, the stigma associated with infertility among Nigerian women is overwhelmingly experienced as these women are described and perceived negatively by members of the community [18,19]. Although there is scarcity of data on infertility in the Northwestern part of Nigeria, of which Zamfara is part, an incidence of 48% is reported [20,21]. Yet, little is known about the experiences of women with infertility in Zamfara. Therefore, this study explored the psychosocial experiences of women with infertility in Zamfara State of Nigeria.

## Materials and methods

### Research design

An exploratory qualitative design was used to explore the psychosocial experiences of women with infertility in the Federal Medical Centre (FMC) in Zamfara state of Nigeria

### Inclusion and exclusion criteria

Women with both primary and secondary infertility, aged between 18 and 49 years, women who could read and write in English, and women who could communicate in Hausa and were receiving treatment for infertility in the FMC were purposively sampled. Women with infertility who were receiving treatment for clinical depression were excluded.

### Ethical approval

The Institutional Review Board of the Noguchi Memorial Institute for Medical Research in the University of Ghana reviewed the proposal and approved the study (IRB 0000908) and permission was sought from the Federal Medical Centre for data collection.

### Tool for data collection

A semi-structured interview guide was designed based on the objectives of the study and used for all interviews. The interview guide comprised of two parts. Section A focused on the socio-demographic characteristics of the participants. Section B comprised of the main questions on their psychosocial experiences of infertility. For content validation, the interview guide was reviewed by qualitative research experts and pre-tested prior to the interviews. The interview guide is attached as appendix A.

### Data collection

Data were collected between February and April 2014. On each day of data collection, the researchers and a trained research assistant met with each client at a venue chosen by the client. After the content of the consent form was fully explained to the participants individually, each participant signed two consent forms; one for the participant and the other for the researchers. The research assistant who was fluent in both English and Hausa was trained to interview and transcribe data from participants who could only communicate in Hausa. Each participant was interviewed for 45 minutes and audiotaped with permission from the participants and transcribed verbatim for content analysis.

During each interview, we paid more attention to the richness of the data rather than the sample size [22]. At the end of each data collection day, the researchers transcribed each audiotape verbatim before the next interview. Transcribing each interview before the next one was necessary to ensure the richness of the data and identity relevant areas for probing in the next interview. Therefore, the interviews and data transcription were concurrent, which made it

easier for the detection of new and redundant information. By the 11[th] interview, no new information was noticed in the interviews. However, one more participant was interviewed to ensure that all the necessary data were gathered for analysis.

### Data analysis

A three-member team conducted data coding. Data coding involved collating together important phrases and sentences that were of relevance to the purpose of the study. These statements were highlighted and assigned a label or code. The coded passages were then compared for similarities and differences. Codes that had common elements were grouped to form main themes and sub-themes. The themes and sub-themes were continually revised for clarity and appropriateness. In all, two main themes (psychological and social experiences) and eight sub-themes (anxiety, depression, stress, isolation, stigma, pressure, marital problems, and support) emerged from the data.

## Results

### Demographic characteristics

The participants were all females aged between 22 and 45 years. All the participants were married, Muslims by religion and the majority were Hausa by tribe. One participant was a registered midwifery tutor, one had a National Certificate of Education, one had primary school certificate, three had secondary school certificates and the remaining six were illiterate. Ten of the participants were in polygamous marriages and two were in monogamous marriages. Half of the participants had primary infertility and equal number had secondary infertility. Four of the participants had one child each, one had three children (but still regarded herself as infertile) and all the others had no children. Five of the participants were engaged in small-scale businesses and five were housewives.

## Psychological experiences

A major theme identified on the basis of the women's experiences was psychological. Women with infertility in Zamfara reported lots of psychological problems in the form of anxiety, depression and stress.

### Anxiety

These women expressed their feelings of anxiety in various ways. Vast majority of the women described their situation as worrying, too much thinking, being in doubt, and fear. Furthermore, some of the women described their feeling of anxiety in the form of hallucinations, as indicated by the following verbatim quote;

> *If I sit alone I think too much*; *sometimes I can`t hear someone talking to me until when I am touched.*

Other participants described their source and level of anxiety as follows;

> *Up till now I am speaking to you, sometimes if I sit alone I start to think about things, that my younger ones have about 5–6 children but me the most senior child of the family, I don't have any.*

> *In fact there is no single day I will not think of the situation that I am facing. I imagine if this is that how I am going to end?*

These narrations were similar among the women's descriptions of their anxiety, suggesting that anxiety is an inevitable part of the psychological experience of infertility among these women.

## Depression

The women were found to suffer depression as a result of their infertility. They described being depressed in different ways including sadness, lack of happiness, restlessness, anger and loss of interest in social communication and interaction. In terms of sadness, many participants similarly expressed the following;

*Sometimes, people ask me why I am looking sad? Others ask if there is there any problem? So even in my matrimonial home I am always looking very sad.*

The women, as indicated by the following quotes, also described lack of happiness as a sign of depression;

*Actually, I can feel deep in me that I am not happy because I spent many years in my marriage without a child.*

*I am never happy because people are saying I am filling their toilets with big stools but no children. How will one be happy in such situations? In that case you cannot do anything to satisfy them because of your inability to get pregnant.*

In terms of restlessness and anger, majority of the women similarly narrated the following;

*Because of this situation I always react negatively with anger towards people.*

*My husband`s relatives have been saying a lot of things. So I have no choice but to be angry.*
Furthermore, a participant reported that the situation at home and in society is such that anger is inevitable;

*Whether I am at home or outside, I am always angry without a cause and sometimes people ask me questions that are annoying and so I get angry.*

These findings suggest that these women unconsciously experienced depression in different forms.

## Stress

The stress symptoms described by the women were in the form of crying, forgetfulness, disturbances, lack of sleep, palpitation and loss of libido. All the women attributed these experiences to abuses from their mother in-laws, husbands, family, and the society as indicated by the following quotes:

*'After being abused by people, I shed a lot of tears when I enter my room.*

*It is very difficult to spend a week without shedding tears'.*

Some of the women reported instances when they had to hide in their rooms just to cry;

*'I hide myself in my room and cry, when I come out from the room people will see the sign of worries on my face and ask what is happening and I tell them that I am not feeling fine'.*

Some of the women described their stress in the form of forgetfulness:

*'I became easily forgetful, I enter into my room to pick something and I will just forget what I am there to do. I am not old enough to develop that'.*

Some women on the other hand, described their experience of stress as being disturbed;

*'I am disturbed because every woman wants to give birth to a child just as the way she was born by her mother'.*

*'I am disturbed because my mother in-law has been complaining why I am not pregnant? I explained to her but she can`t consider my explanations'.*

Furthermore, the inability to eat, sleep and frequent memories of the situation were described by some women as their experience of stress as stated:

*'Sometimes I cannot not eat, I cannot even be able to sleep'.*

*'If a small negative thing happens to me, it will remind me of a lot of things I passed through'.*

*'Sometimes I lack interest in sexual intercourse simply because I saw the result of the past period which ended without pregnancy'.*

These narratives testify to the women's gruesome experiences of infertility-related stress.

## Social experiences

Another major theme that emerged was the social experiences of these women, which were linked to the social aspects of their lives. These social experiences include isolation, which could be either self or social, social stigma, social pressure, and marital problems.

### Isolation

The women reported two forms of isolation. These were self-isolation and social isolation. Self-isolation was a choice made by the women themselves as a way of life based on their experiences:

*'Even if they gather I don't put myself among them. I will be doing my things alone'.*

The women gave various reasons for isolating themselves from the society in order to avoid statements that may affect them negatively:

*'I avoid taking part in their conversation because I don't want someone to say a word that will hurt me or affect me negatively. In such instances, I may respond and that can lead to a conflict'.*

Others isolated themselves in order to avoid mockery:

*'Though you think more while in a room alone, I prefer that sometimes because it makes you avoid people's mockery'.*

Social isolation was as a result of the society isolating the women from social activities and decision-making. According to many of the women, people isolated them whenever they had discussions about children:

*'There was a time when they were going to do something that has to do with children, they said leave her, she doesn't have experience on that, she will not give us the experience that we require'.*

Some of the women reported that their opinion was not sought anytime their families had to take crucial decisions:

*'I don't know why in so many instances I will just hear things done in my house without consulting me. For instance, when my husband was getting married, I didn't know anything about it, just suddenly I was told to come and take lefe [*marital gift*] to the lady he was going to marry'.*

Some of the women reported social isolation in the form of people preventing children from staying in the surroundings of these women:

*'Whenever I had a conflict with my rival she will call her children back to her room and tell them to avoid my sight'.*

*'Most of the time, they don't allow their children to stay in my house, they always call them back'.*

## Social stigma

The women described two dimensions of stigma. These were actual stigma experienced and perceived stigma. For actual sigma, the women were branded as women who control their fertility and they found this stigmatizing. They reported situations in which people described them as users of family planning, lovers of high education, being barren/infertile and aggressive. Perceived stigma was described in the form of mockery. Actual stigma was experienced in various ways as narrated below:

*'People were telling me to stop taking family planning, which means I must be using family planning that is why I am not pregnant'.*

*'My neighbours think that I am having fertility problem because I am using family planning or I used them before'.*

Almost all the women narrated their experience of actual stigma as people perceiving that they are women who intentionally do not want to get pregnant:

*'Some are thinking that since I am a health personnel I have something that I am doing to prevent myself from getting pregnant'*

*'They think that there is something I am doing secretly, they think I am the cause'.*

Some of the women were also stigmatized for being aggressive because of their inability to conceive:

*'People are saying that I am aggressive, sometimes they even say that I am inhuman'.*

Women who had higher education reported being stigmatized as infertile because they did not marry early:

*'They say that, I have infertility because I spent many years in school without marriage. So my high education is used against me'.*

## Social pressure

The women were faced with challenges that imposed pressure on them because of their infertility. The pressure was reported as actual or perceived pressure.

Actual pressure was described as the one imposed directly on the women by the family, husband or society. Majority of the women reported actual pressure as one of the problems they faced:

*'My husband shows that my rival is more important than me because she gets pregnant and delivers which I cannot do'.*

*'Whenever my rival delivers, my husband changes towards me and gives her more attention than me'.*

Some of the women endured pressure from their husbands because the husbands expressed their desire to marry another wife:

*'My husband found it to be his hobby to be telling me that he will marry another wife, he always threatens me with such offensive words'.*

Some had memories of their husbands attempting to send them away because of their inability to get pregnant;

*'It has been hurting me whenever I remembered that some years back, my husband sent me out of his house because I couldn't get pregnant!'*

Because the majority of the women were married and living in polygamous families, they reported having pressures from their rivals:

*'My rival despite the good things I have been doing for her, there were times when she said I was jealous of her pregnancy'.*

Pressure from mother in-laws was reported by the women and described in different ways. For instance, some mother in-laws threatened divorce as follows:

*'My mother in-law asked my husband to divorce me because of my inability to get pregnant'.*

In summary, these narrations of social pressures indicate that infertility is indeed a social problem apart from being medical.

## Marital problems

Marital problems are the misunderstandings that occurred between the husbands and their wives as a result of infertility. In polygamous families, marital problems also involved problems with rivals. Marital problems reported by the women included divorce, marrying another woman, and conflicts. Few women who were divorced just before they were interviewed had these to share:

> 'Although I couldn't know the reason for the divorce, I heard that he said the reason why he divorced me was because his relatives mocked at him'. 'They mocked at him because he married me for 2–3 years but I couldn't conceive. My mother in-law forced my husband to divorce me'.

The women reported that their husbands married other women because of their infertility. All the women described either their mother in-laws or husband's relative as the driving force behind marrying other women:

> 'They got a lady in my neighbourhood and made him to marry her, you see, the infertility is the root of all these'.

> 'Actually, when they came to understand that I have a problem, his parents told him to get another wife and he did so'.

Conflict was another marital problem described by the women. The women reported having misunderstandings with their husbands, rivals, and husbands' relatives as a result of their inability to get pregnant. However, some women reported being resistant to ridicule, which also led to conflicts;

> 'I don't accept what they do to me any longer. My husband and my rival call me infertile who fills their toilet with stools, so I fight with them any time they make such utterances to me'.

Consequently, it is worth noting that infertility appears to be a recipe for many marital problems.

## Discussion

Women with infertility in Zamfara reported numerous psychosocial problems of infertility such as anxiety, depression and stress. The findings indicated that anxiety frequently affects women with infertility and is manifested in various ways as reported in the literature [13]. However, a quantitative assessment of the levels of anxiety is necessary among women in Zamfara. The stress symptoms described by these women were forgetfulness, lack of sleep, palpitations, and loss of libido. These experiences of stress may have resulted from a multitude of factors such as the desire to attain motherhood, the strain imposed on marital relationships as well as societal pressure. More importantly, the feeling of loneliness, verbal abuse and social isolation were also disturbing problems reported. These findings suggest that these women deserve some sort of psychosocial attention to improve their wellbeing.

The women's psychological experiences were linked to their social life experiences. Women in this study were found to suffer social isolation as found in other parts of Africa [13,14]. However, the present study reported self-isolation used by these women to avoid conflicts. Given the fact that these women experienced various forms of mockery in the family and society, their self-isolation may be justifiable but inappropriate for their social health.

Furthermore, similar to findings in Ghana [17], actual stigma was reported in this study, where the women were branded as controllers of their fertility and aggressive women. These kinds of stigma related descriptions, if ingrained in the community may be dangerous to the uptake of contraceptive use among women in Zamfara.

This study found that the women were confronted with challenges from mother in-laws who had a lot of negative influences on their psychosocial experiences. These challenges could possibly be attributed to the significance of childbearing among the people of Zamfara, which compounded their negative experiences. As found in the present study, infertility has been reported to cause marital problems among couples [12,13]. One of the major marital problems found in this study was divorce, where some of the women were either divorced or threatened with divorce, suggesting that women with infertility have a greater possibility of unstable marriages. Furthermore, for those who may not be divorced, they stand the chance of being coerced into polygamy because of their husbands desire to marry second wives.

## Conclusion

This study is the first of its kind conducted in Zamfara State to explore the psychosocial experiences of women with infertility. The findings suggest that women with infertility in Zamfara State experience a lot of psychosocial problems that are associated with being infertile. It may therefore, be deemed appropriate to envisage infertility as a psychosocial health problem rather than just a pure medical problem. It is important to consider cognitive behavioural therapy for infertility because mitigation of the psychosocial problems may significantly improve the biological problems of infertility.

## Appendix A
### INTERVIEW GUIDE

Title: "My Mother in-law forced my husband to divorce me": Experiences of Women with infertility in Zamfara State of Nigeria.

**Section A**
Personal Data:
Please tell me about yourself
Age. Occupation. Religion. Tribe. Education.
**Section B**
**Psychological experiences**

1. Can you share with me how you felt when you were told that, you have infertility?

2. As a married woman with this condition how have you been feeling?

3. What reminds you of this situation?

4. Tell me how this infertility affects your thinking?

5. How do you perceive life in this situation?

   **Social Experiences**

1. Can you kindly share with me the situation in your matrimonial home after you were told you have difficulty getting pregnant?

2. Considering our culture, which insist that every woman must have children, how are you dealing with this situation?

3. From your experiences, how does society look at you?

4. From your understanding of the situation, how will you compare your position in the society before and after the diagnosis?

5. Can you please describe how you relate with people before and after the diagnosis?

## Supporting information

**S1 Transcript. Respondent 1.**
(DOCX)

**S2 Transcript. Respondent 2.**
(DOCX)

**S3 Transcript. Respondent 3.**
(DOCX)

**S4 Transcript. Respondent 4.**
(DOCX)

**S5 Transcript. Respondent 5.**
(DOCX)

**S6 Transcript. Respondent 6.**
(DOCX)

**S7 Transcript. Respondent 7.**
(DOCX)

**S8 Transcript. Respondent 8.**
(DOCX)

**S9 Transcript. Respondent 9.**
(DOCX)

**S10 Transcript. Respondent 10.**
(DOCX)

**S11 Transcript. Respondent 11.**
(DOCX)

**S12 Transcript. Respondent 12.**
(DOCX)

## Acknowledgments

We are grateful to the staff and patients of the Federal Medical Centre of Zamfara State, Nigeria, for their support.

## Author Contributions

**Conceptualization:** Florence Naab.

**Data curation:** Yakubu Lawali.

**Formal analysis:** Florence Naab.

**Methodology:** Florence Naab, Yakubu Lawali.

**Supervision:** Florence Naab, Ernestina S. Donkor.

**Validation:** Florence Naab, Ernestina S. Donkor.

**Visualization:** Florence Naab.

**Writing – original draft:** Florence Naab.

**Writing – review & editing:** Florence Naab.

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
