## [Decision Letter · Decision Letter 0]

27 Jul 2019

PONE-D-19-15263

“My mother in-law forced my husband to divorce me”: experiences of women with infertility in Zamfara State of Nigeria.

PLOS ONE

Dear DR. Naab,

Thank you for submitting your manuscript to PLOS ONE. After careful consideration, we feel that it has merit but does not fully meet PLOS ONE’s publication criteria as it currently stands. Therefore, we invite you to submit a revised version of the manuscript that addresses the points raised during the review process.

We would appreciate receiving your revised manuscript by Sep 10 2019 11:59PM. To enhance the reproducibility of your results, we recommend that if applicable you deposit your laboratory protocols in protocols.io, where a protocol can be assigned its own identifier (DOI) such that it can be cited independently in the future. For instructions see: http://journals.plos.org/plosone/s/submission-guidelines#loc-laboratory-protocols

We look forward to receiving your revised manuscript.

Kind regards,

Susan Bartels, MD, MPH, FRCPC

Academic Editor

PLOS ONE

Journal Requirements:

1. Thank you for including your ethics statement in your manuscript:

"The Institutional Review Board of the Noguchi Memorial Institute for Medical

Research in the University of Ghana reviewed the proposal and provided ethical

clearance (IRB 0000908) for the research and permission was sought from the

Committee of Federal Medical Centre at Gusau, Nigeria for data collection.

Please amend your current ethics statement to confirm that your named institutional review board or ethics committee specifically approved this study.

2.

We note that you have indicated that data from this study are available upon request. PLOS only allows data to be available upon request if there are legal or ethical restrictions on sharing data publicly. For more information on unacceptable data access restrictions, please see http://journals.plos.org/plosone/s/data-availability#loc-unacceptable-data-access-restrictions.

3. Please amend your manuscript to include your abstract after the title page.

Additional Editor Comments (if provided):

Thank you for your interesting work on this important topic. The manuscript has been independently reviewed by two scholars who have made recommendations aimed at improving the paper.

Reviewers' comments:

Reviewer's Responses to Questions

**Comments to the Author**

1. Is the manuscript technically sound, and do the data support the conclusions?

Reviewer #1: Yes

Reviewer #2: Yes

2. Has the statistical analysis been performed appropriately and rigorously? 

Reviewer #1: Yes

Reviewer #2: N/A

3. Have the authors made all data underlying the findings in their manuscript fully available?

Reviewer #1: No

Reviewer #2: Yes

4. Is the manuscript presented in an intelligible fashion and written in standard English?

Reviewer #1: Yes

Reviewer #2: Yes

5. Review Comments to the Author

Reviewer #1: * The study is an interesting one that revealed the experiences of women in the Northern Nigeria, however, the title

Quote 'My Mother in-law forced my husband to divorce me' was not reflected in the body of the work.

* The 10th line in the first paragraph, 'in the absence of a conception' ought not be there.

* The qualitative nature of the study brought out the originality of the study, however the data/transcripts would have been

provided for review.

* The Study was conducted in Nigeria, why was the ethical clearance obtained in Ghana?

* It was stated in the methodology that women who could read and write in English and communicate in Hausa between the

ages of 18-49 were purposively sampled, the demographic result presented was not the same.

Reviewer #2: This paper, based on a small qualitative study clearly demonstrates the social and psychological problems 12 infertile women in Zamfara experience. It adds to the body of literature showing the poor social status and predicaments of infertile women – which may contribute to putting infertility higher on the agenda of sexual and reproductive health programmes and policies. The study methodology and findings are neatly presented and the quotations well used to illustrate the arguments.

I have a few issues that should be added and a few suggestions to make the paper stronger.

The paper is based on very few interviews and could be much stronger with putting the findings in socio-cultural contexts and further analysis of data, as suggested below.

• In the introduction, refer to studies in Nigeria on the socio-cultural contexts of infertility (see for instance Okonofua, Pearce, Koster-Oyekan) that explain the higher stigma and negative social consequences for the wife than the husband if a coupe is infertile. This contexts relate to the unequal gender relations, patrilineal societies and the social and spiritual importance of having children, for the wife, the couple, and the family (in-law). Only having a child will give a Nigerian wife a status in her family-in-law. In your discussion you do refer to the importance of child bearing – but this should have been already referred to in the introduction. It is a missed chance that you refer to so few studies in Nigeria, while referring to many studies in other African countries and even Turkey.

• With the quotes: give background of the woman talking (for instance: 25 yrs, married, 1 child). In the quotes, put your explanations in square brackets – for instance when they use the word rival [co-wife]. And as much as possible (as far as you know) mention other relevant background of the woman to understand her experiences; such as: time with problems to conceive, first/second marriage, living situation (with in-laws?), time under treatment, diagnosis and prognosis etc. These background variables will influence the social and psychological problems the woman experiences. You could analyse see some trends.

Needed additions/corrections/explanations:

• In introduction make the objectives and relevance of the study more explicit (in tool for data collections you refer to objectives).

• Elaborate on selection of study participants – among those receiving treatment. Why did you only select women – were there no infertile men? (See presented interview guide – it mentions sex under personal data)

• Page 6 – Social Stigma, third line. Branded as ‘’controllers’’ Is this the word they used? Can you explain what they mean?

• Check References, formatting (capitals with 15 and 19?); 18 and 19 – a different Upkong?

• Be more nuanced with recommending psycho-social counselling of women as the solution to improve infertile women’s well-being and solving biological problems. Put a reservation referring to unequal gender relations and importance child bearing for status of women.

6. PLOS authors have the option to publish the peer review history of their article (what does this mean?). If published, this will include your full peer review and any attached files.

Reviewer #1: Yes: Oyetunji-Alemede, Catherine Olajumoke

Reviewer #2: No

---

## [Author Response · Author response to Decision Letter 0]

9 Oct 2019

Responses to reviewers comments have been submitted in a rebuttal letter.

---

## [Editor Report · Decision Letter 1]

30 Oct 2019

“My mother in-law forced my husband to divorce me”: experiences of women with infertility in Zamfara State of Nigeria.

PONE-D-19-15263R1

Dear Dr. Naab,

We are pleased to inform you that your manuscript has been judged scientifically suitable for publication and will be formally accepted for publication once it complies with all outstanding technical requirements.

With kind regards,

Susan Bartels, MD, MPH, FRCPC

Academic Editor

PLOS ONE
---

## [Editor Report · Acceptance letter]

11 Dec 2019

PONE-D-19-15263R1 

“My mother in-law forced my husband to divorce me”: experiences of women with infertility in Zamfara State of Nigeria. 

Dear Dr. Naab:

I am pleased to inform you that your manuscript has been deemed suitable for publication in PLOS ONE. Congratulations! Your manuscript is now with our production department. 

With kind regards,

on behalf of

Dr. Susan A. Bartels 

Academic Editor

PLOS ONE